# The Effect of Annealing Temperature on the Synthesis of Nickel Ferrite Films as High-Capacity Anode Materials for Lithium Ion Batteries

**DOI:** 10.3390/nano11123238

**Published:** 2021-11-29

**Authors:** Mansoo Choi, Sung-Joo Shim, Yang-Il Jung, Hyun-Soo Kim, Bum-Kyoung Seo

**Affiliations:** 1Decommissioning Technology Research Division, Korea Atomic Energy Research Institute, Daejeon 34057, Korea; bumja@kaeri.re.kr; 2Battery Research Center, Korea Electrotechnology Research Institute, Changwon 51543, Korea; sjsim@keri.re.kr (S.-J.S.); hskim@keri.re.kr (H.-S.K.); 3LWR Fuel Technology Division, Korea Atomic Energy Research Institute, Daejeon 34057, Korea; yijung@kaeri.re.kr

**Keywords:** anode, thermal treatment, nickel ferrites, li-ion batteries

## Abstract

Anode materials providing a high specific capacity with a high cycling performance are one of the key parameters for lithium ion batteries’ (LIBs) applications. Herein, a high-capacity NiFe_2_O_4_(NFO) film anode is prepared by E-beam evaporation, and the effect of the heat treatment is studied on the microstructure and electrochemical properties of LIBs. The NiFe_2_O_4_ film annealed at 800 °C (NFO-800) showed a highly crystallized structure and different surface morphologies when compared to the electrode annealed at a lower temperature (NFO-600, NFO-700). In the electrochemical measurements, the high specific capacity (1804 mA g^−1^) and capacity retention ratio (95%) after 100 cycles were also achieved by the NFO-800 electrode. The main reason for the good electrochemical performance of the NFO-800 electrode is a high structure integrity, which could improve the cycle stability with a high discharge capacity. The NiFe_2_O_4_ electrode with an annealing process could be further proposed as an alternative ferrite material.

## 1. Introduction

Ever growing concerns of energy consumption are one of the important issues in modern society [1]. In order to meet an increasing energy demand, the Li-ion battery (LIB) has been considered as a potential candidate for electric vehicles and energy storage systems, which should have a high power and energy densities [2]. To have a high performance in an LIB, it is necessary to design optimized anode materials with a high capacity and long cycle stability. These requirements can be fulfilled with an anode electrode development with a higher specific capacity than commercial anode material such as graphite (372 mAh g^−1^). The transition metal oxide that has a conversion reaction shows higher specific capacities when compared to intercalation compounds. For instance, ferrites with the general formula of an MFe_2_O_4_ (M = bivalent cation) spinel oxide, are considered promising anode materials because of their high specific capacities, natural abundance, and low cost. Among the ferrite materials, NiFe_2_O_4_ is an inverse spinel structure where Ni^2+^ and half of the Fe^3+^ cations occupy the octahedral sites and the remaining Fe^3+^ are on the tetrahedral site. Moreover, NiFe_2_O_4_ delivers its theoretical capacity of 915 mAh g^−1^, which can electrochemically react with 8 mol of Li. Despite of advantages of NiFe_2_O_4_, the volume expansion of the electrode during the charge-discharge process resulted in a poor cycle performance [3]. As a result, this prevents them from being practical anode materials for LIBs (lithium ion batteries’). Several studies discussed anode materials with a higher specific capacity, rate capability, and cycle performance than graphite [3,4,5,6]. Various strategies have been proposed to overcome the above obstacles related to the NiFe_2_O_4_ electrode by employing nanostructures such as nanosheet, nanowire, and nanoparticles [7,8,9]. The nanostructured electrode is favorable to an outstanding electrochemical performance owing to the short distance of Li ion diffusion and large surface area [9]. Another strategy has been that nanostructured carbon such as a carbon nanotube or graphene has been used as a support for the NiFe_2_O_4_ electrode [10,11]. The nanocarbon materials could not only improve the electronic conductivity but also be served as buffering media against the volumetric expansion of the electrode. It has been reported that the deposition method could be a favorable method to prepare a (Ti, Fe)-alloyed Si thin film anode with an accurate stoichiometric composition of materials [12]. Although LiMn_2_O_4_ (LMO) can be synthesized at room temperature in tetragonal and cubic forms, it can be synthesized at high temperatures, and the influence of the annealing temperature on the physical and electrochemical properties of the LMO thin film is investigated as an anode material [13,14,15].

In this study, the NiFe_2_O_4_ film as an anode electrode was prepared by a one-step synthesis using the E-beam evaporation equipment. The as-prepared NiFe_2_O_4_ film was followed by a heat treatment and directly used as an anode electrode for LIBs without further process such as adding conductive materials or binder, or conducting pressure roast processes. The effects of the heat temperature of the NiFe_2_O_4_ film were systematically investigated, and the structural and electrochemical properties will be discussed in details.

## 2. Materials and Methods

NiFe_2_O_4_ film was coated on stainless foil (SUS 304, thickness 25 μm) prepared by an electron beam evaporation system. Before the deposition process, the chamber was evacuated down to 3 × 10^−6^ Torr, and the surface of the stainless foil was etched by Ar ions for 5 min before deposition to remove any oxide layer. A NiFe_2_O_4_ pellet (purity 99.9%, from KOJUNDO chemical laboratory Co. LTD, Saitama, Janpan) was used as the target, and the deposition rate (0.5 Å/s) was monitored by a thickness sensor. For the investigation of the heating effect, the NiFe_2_O_4_ film was annealed at 600, 700, 800 °C, for 1 h in Ar atmosphere.

The morphology of the NiFe_2_O_4_ film surface were examined by field emission scanning electron microscopy, FESEM, (S-4800, Hitachi, Tokyo, Janpan), and a high resolution transmission electron microscope, HRTEM, (JEOL-2100F, JEOL, Tokyo, Japan) an elemental mapping analysis of the NiFe_2_O_4_ sample was also performed with energy dispersive spectroscopy. For the cross-sectional image, we prepared the sample by using focused ion beam-scanning electron microscopy (FID-SEM, HITACHI, Tokyo JAPAN) with Ga ion sputtering. To prevent a morphological change by the ion beam, the Pt coating was applied on the surface of the electrode. The thin film X-ray diffraction (X-pert PRO MRD, Philips, Washington, DC, USA, Cu Kα radiation (λ = 1.5406 Å)) was carried out 2*θ* from 10° and 90° at a scan rate of 0.01°. A Raman measurement (NTEGRA SPECTRA, NT-MDT, Tempe, AZ, USA) was conducted with an Ar laser excitation source emitting at a wavelength of 514 nm. The elemental composition and depth profile were characterized by the X-ray photoelectron spectroscopy system. The X-ray source was used with monochromatic Al Kα radiation (1486.6 eV), and the spectra was calibrated using C 1s (BE = 284.8 eV) of hydrocarbon. 

The electrochemical performance of the NFO anode electrodes was evaluated using 2032 coin cells assembled in a dry room. The Li foil and polypropylene 2400 (Celgard, Charlotte, NC, USA) were used as a counter and reference, and as a separator, respectively. The electrolyte was 1.3 M LiPF6 in ethylene carbonate (EC) and diethyl carbonate (DEC) (3:7 in volume) (Soulbrain, Northville, MI, USA). Galvanostatic charge-discharge and the cyclic voltammetry (CV) tests were performed within a potential range of 0.005–3.0 V (vs. Li^+^/Li) with an applied current density of 0.1 C (1 C = 915 mA g^−1^) and at a scan rate of 0.5 mV s^−1^, respectively. All the electrochemical measurements were carried out at room temperature using a battery test system (Battery Analyzer, Won-A tech, Daejeon, Korea). The mass loading of NFO on the stainless steel according to the density of NFO, and the average thickness of NFO from the cross-section SEM images, is about 0.3 mg and 500 nm. The cyclic voltammetry (CV) was recorded in the potential range of 0.005–3.0 V at a scan rate of 0.5 mV s^−1^. All the electrochemical measurements were carried out at room temperature. 

## 3. Results

The crystal structure of NiFe_2_O_4_ film according to the heat temperature was determined by X-ray diffraction (XRD, X-pert PRO MRD, Philips, Washington, DC, USA). Figure 1 shows the XRD patterns of the NFO-600, NFO-700, and NFO-800 electrodes. Obviously, the crystallinity of NFO film electrodes depend on the heat treatment temperature, and they are shown in Figure 1, where it can be seen that with an increase of the temperature from 600 to 800 °C the peaks become sharper, resulting in a higher crystallinity of the NiFe_2_O_4_ materials. 

As shown in Figure 1, all samples have shown diffraction peaks at a similar 2*θ* degree when compared with the NiFe_2_O_4_ target peak, corresponding to the (220), (311), (222), (400), (511), and (440) crystal planes of the NiFe_2_O_4_ (JCPDS No. 54-0964) [16,17,18]. These diffraction peaks confirmed the presence of single-phase NiFe_2_O_4_ with a face-centered cubic and F*d3m* space group. The peak intensity becomes sharper when the annealing temperature is increased from 600 to 800 °C, resulting in a higher crystallinity for NFO-800 than NFO-600. The crystallite sizes of the NFO-600, NFO-700, and NFO-800 electrodes are calculated as 17, 19, and 22 nm, respectively, using Scherrer’s equation. Peaks related to any other phase or impurities were not observed. The peak at 45° corresponds to the stainless foil (denoted by a star), which is similar to previous studies [19,20].

Figure 2a–c shows the surface morphologies of the NFO-600, NFO-700, and NFO-800 electrodes. As can be seen in Figure 2a,b, the NiFe_2_O_4_ film shows numerous streaks based on the mother substrate, without any cracking, and a few droplets are also observed. These particles are ascribed to the incomplete elimination of target splashing during deposition [17]. After the heat treatment at 800 °C, however, the surface morphology is very different from electrodes heated at a lower temperature. The roughness and porosity of the surface is observed, evidencing the effect of the annealing temperature. This observation of a roughened surface suggests that the film might be undergoing a change of orientation. It is reported that the heat treatment of stainless steel changes its chemical composition and the thickness of the passive film [21]. Thus, the surface morphologies are dependent on the heat temperature. It is anticipated that the roughened surface of the NiFe_2_O_4_ electrode could lead to a low resistance interface of the electrode. Besides, the thickness of the electrode on glass was about 500 nm (Figure 2d), and the conformal NiFe_2_O_4_ film was clearly observed in the cross-sectional image to be without any splits.

Figure 3 shows the cross-sectional TEM image with energy dispersive X-ray spectrum (EDS) elemental maps of the NFO-800 electrode. The clearly defined NiFe_2_O_4_ electrode and SUS substrate were observed. The EDS elemental mapping analysis was also conducted in order to observe the elemental composition distribution at the NiFe_2_O_4_/interface between the oxide layers and stainless steel. As shown in Figure 3, Ni, Fe, and O were distributed in the outer layer, which indicated that Cr did not exist in the NiFe_2_O_4_ electrode. However, the Fe element was depleted in the interface between the oxide and substrate. In contrast to Fe, the Cr element becomes denser at the interface. This indicates that the Cr ions are diffused from the substrate to the outer oxide layer, which is composed of chromium and oxygen represented as chromium oxide. It is reported that the duplex oxide layer is formed at a high temperature with the inner and outer layer [22].

The outer layer is grown by the out-diffusion of metal ions from the substrate. Moreover, the stainless steel containing sufficient chromium established a protective chromium rich oxide layer as a Cr_2_O_3_ and Fe-Cr spinel structure during oxidation at a high temperature [23]. Consequently, the NiFe_2_O_4_ electrode was formed by NiFe_2_O_4_ (outer layer)/Cr_2_O_3_ (transition layer), Fe-Cr spinel, and stainless steel (substrate) from the outer to inner oxide layer. In a previous study, low density regions were observed at the interface between NiFe_2_O_4_ and the substrate due to the high density of this film and the different temperature expansion coefficients (10.3 × 10^−6^ K^−1^ of NiFe_2_O_4_ and 16 × 10^−6^ K^−1^ of stainless foil) [17]. In this study, however, the NiFe_2_O_4_ electrode annealed at a high temperature has been formed without any crack or low-density regions. We have assumed that the internal phase (Cr_2_O_3_ and Fe-Cr spinel) could be beneficial for achieving an enhanced electrochemical performance. Figure 4 shows the TEM cross-sectional image of the NFO-800 and associated selected area electron diffraction (SAED) patterns of the inner layer and outer layer of the NiFe_2_O_4_ electrode. As shown in Figure 4, the NiFe_2_O_4_ electrode was formed as a double-layer structure, which is shown in Figure 3. The inner and outer layer of the NiFe_2_O_4_ electrode show spinel SAED patterns. The d-values calculated from the concentric rings of both layers match well with (220), (311), and (400), indicating the spinel-type structure of the NiFe_2_O_4_. The SAED patterns from the inner layer show a spot pattern, whereas the outer layer (NiFe_2_O_4_) indicates a ring pattern. This difference could be attributed to the amount of Cr in the spinel structure.

Figure 5 shows the typical Raman spectra of the NiFe_2_O_4_ electrode annealed at 800 °C. As can be seen in Figure 5, the Raman spectra showed the inverse spinel structure of NiFe_2_O_4_, which is consistent with previous reports [24,25]. It is reported that the NiFe_2_O_4_ spinel structure group can be described with the space group of F*d3m* (No. 227), and the factor group theoretical calculations result in five Raman active bands, namely *A_1g_* + *E_g_* + 3*T_2g_* [25]. In this study, the Raman spectra have shown the crystallized NiFe_2_O_4_ electrode, which confirms the XRD result. Any peaks related to the NiFe_2_O_4_ were not observed on the SUS substrate.

The XPS depth profiling was carried out to observe the chemical concentration of iron, nickel, and chromium with Ar^+^ ion beam sputtering. Figure 6 shows the XPS depth profiles of NiFe_2_O_4_ according to the annealing temperature. As shown in Figure 6, in the NiFe_2_O_4_ region, Ar^+^ etching extending up to 4000 s was clearly observed, and the thickness of the NiFe_2_O_4_ electrode was similar before and after annealing except for the NFO-800 electrode. Before the heat treatment of NiFe_2_O_4_ electrodes, the chemical ratio of Fe:Ni was irregularly observed, but the composition ratio of Fe:Ni was consistently close to the theoretical composition of the NiFe_2_O_4_ electrode after the heat treatment. This indicates that the NiFe_2_O_4_ electrodes may be undergoing a change of orientation during annealing. The Cr element was much more increased at the interface layer between NiFe_2_O_4_ and the SUS substrate, which was consistent with the EDS mapping result. However, the Fe and Ni of the NFO-800 electrode showed different chemical compositions when compared with a lower annealing temperature. The Cr elements were outwardly diffused at 800 °C from substrate, which resulted in more migration of Fe and Ni from the NiFe_2_O_4_ electrode to the interface. From the results of the XPS depth profile, the stainless steel annealed at a high temperature established a protective chromium rich oxide (Cr_2_O_3_) and Fe-Cr spinel structure in the interface layer [23].

Figure 7 shows the charge-discharge profiles of the NiFe_2_O_4_ electrode at 0.1 C in the range of 0.005–3.0 V, with all of the electrodes showing typical voltage profiles of the discharging (lithiation) and charging process (delithiation). In the first discharge process, a long-range plateau was observed around 0.75 V, which corresponded to the reduction of Fe^3+^/Ni^2+^ during Li insertion. This constant region from 0.7 to 0.5 V showed a specific capacity of more than 1000 mAh g^−1^. This indicates that Ni and Fe has been completely converted to metallic elements and the formation of Li_2_O [26]. In the charge process, the quasi-plateau was also observed at 1.5 V, which resulted in the oxidation of transition metals [26]. A similar charge-discharge process has been observed in other transition metal oxides [27,28]. The second and third cycles almost remained the same profile, which revealed that the NiFe_2_O_4_ electrode became stable and had reversible reactions. The initial discharge capacities of the NFO-600, NFO-700, and NFO-800 electrodes were 1551, 1553, and 1804 mAh g^−1^, and the columbic efficiencies of each sample were 76, 77, and 76%, respectively. The discharge capacity obviously showed the highest capacity at an annealing temperature of 800 °C. The discharge capacities of the NiFe_2_O_4_ electrodes exceed the theoretical capacity of NiFe_2_O_4_ (~915 mAh g^−1^), which is attributed to the decomposition of the electrolyte when the potential is decreased at a low voltage [29]. However, the initial columbic efficiencies (CE) were relatively low. The reason for the low columbic efficiencies is the formation of a solid electrolyte interface film (SEI) film of the electrode and the decomposition of the electrolyte [30]. In the subsequent cycle, the CE was increased to 98%.

In order to investigate the reaction mechanism of NiFe_2_O_4_, the cycle voltammetry (CV) of NFO-800 was carried out at a scan rate of 0.1 mV s^−1^ in the potential range of 0.005–3.0 V and is depicted in Figure 8. In the first cathodic cycle, a large peak at 0.7 V is observed, which is attributed to the reduction reaction of Fe^3+^ and Ni^2+^ to Fe^0^ and Ni^0^, respectively. In the subsequent cycles, the reduction peaks were shifted to 0.8 and 1.1 V, indicating the irreversible reaction in the first cycle. For the reversible cycle, the anodic peak at 1.6 V was related to the oxidation of Fe and Ni to Fe^3+^ and Ni^2+^, and it shifted to 1.9 V in the second cycle due to the change in the ionic environment [31]. The redox mechanism observed during the charge-discharge reaction may occur in the following steps:NiFe_2_O_4_ + 8Li+ + 8e^−^ ↔ Ni + 2Fe + 4Li_2_O
Ni + Li_2_O ↔ NiO + 2Li+ + 2e^−^
2Fe + 3Li_2_O ↔ Fe_2_O_3_ + 6Li^+^ + 6e^−^

In addition, the subsequent cathodic and anodic CV cycles almost overlapped, indicating a stable cycle performance against the Li.

Figure 9 reveals the cycle performance of the NiFe_2_O_4_ electrode according to the annealing temperature at a current density of 91.5 mA g^−1^ for 100 cycles. As shown in Figure 9, the specific discharge capacities of the first cycle of annealed NFO-600, NFO-700, and NFO-800 electrodes were 1551, 1559, and 1804 mA g^−1^, respectively. The CE of the first cycle of three different electrodes was around 76.7, 77.6, and 77%, respectively, which was attributed to the formation of the SEI film. After the first cycle, the CE was retained to more than 99% until 100 cycles, indicating the superior cycle stability of the electrode. The discharge capacities of the NFO-600 and NFO-700 electrodes gradually decreased during 30 cycles and then stabilized at their capacity until 100 cycles. However, the NFO-800 electrodes showed less capacity degradation in the initial cycles and a much higher reversible capacity of 1086 mAh g^−1^ during the cycles. The capacity retention ratio of the NFO-600, NFO-700, and NFO-800 electrodes was 65, 72, and 95% after 100 cycles, respectively. This indicates that the annealing process of the NiFe_2_O_4_ electrodes at a high temperature provides a favorable lithium storage capacity and exceptional cycle stability. The reason for the high capacity retention ratio of NFO-800 was the good structural integrity during the cycling. Figure 10 shows the surface morphologies and cross-section images of the NFO-800 electrode before and after the cycle. Before cycling, the NiFe_2_O_4_ electrode surface was interconnected with distinct NiFe_2_O_4_ particles, indicating a relatively good electron pathway. However, the NiFe_2_O_4_ particles were densely embedded and showed a smooth surface without any cracks or individual particles. Thus, the optimization of the annealing temperature could lead to a high structural integrity of the NiFe_2_O_4_ electrode. The C-rate performance will be carried out when pursuing further research.

Based on the above results, the annealed NiFe_2_O_4_ electrodes not only improve the high crystal structure, but also show better electrochemical properties. The NiFe_2_O_4_ electrode with an interface layer between NiFe_2_O_4_ and the SUS substrate has given a higher discharge capacity than the theoretical NiFe_2_O_4_ anode materials. In particular, the NFO-800 electrodes have maintained their structural integrity, which improves the cycle stability with a high discharge capacity after 100 cycles. Moreover, this study reveals different physical and electrochemical properties of the NiFe_2_O_4_ electrode when compared to a conventional electrode that is based on a copper current collector. Finally, the NiFe_2_O_4_ electrode prepared by an e-beam evaporator followed by an annealing process should be an ideal electrode design with a strong adhesion onto the substrate when compared to other preparation methods.

## 4. Conclusions

In summary, NFO thin film electrodes have been successfully prepared via deposition method, and the annealing effects on the structural and electrochemical properties of NFO electrodes were investigated in a Li-ion battery system. The XRD diffraction demonstrated the high crystallinity of NiFe_2_O_4_, and the interlayer of Cr_2_O_3_ was a beneficial feature that was ascribed to a higher specific capacity after the annealing process. The NFO-800 electrode that showed a charge capacity of about 1100 mAh g^−1^ was stable after 100 cycles at a current density of 91.5 mA g^−1^ (0.1C) and with a high capacity retention ratio of 95%. The NiFe_2_O_4_ electrodes also maintained their reversible capacity of 1100 mAh g^−1^ after 100 cycles. The unique properties of the NiFe_2_O_4_ electrode with the annealing process could be further proposed for alternative ferrite materials, which is applicable for anode electrodes for LIB.

## Figures and Tables

**Figure 1 nanomaterials-11-03238-f001:**
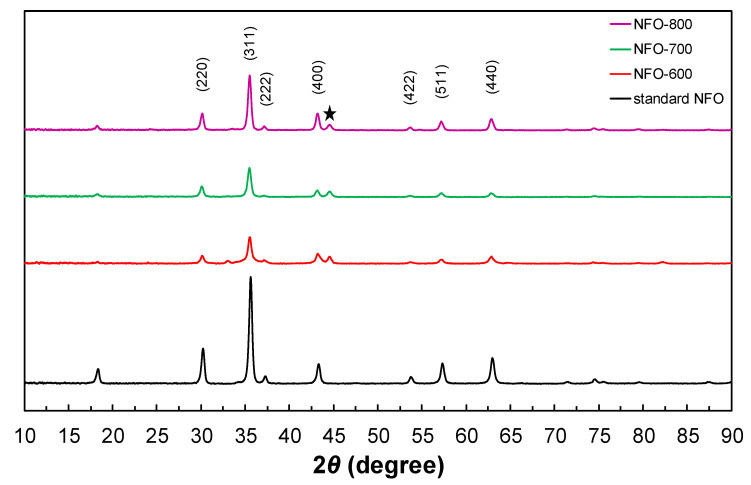
X-ray diffraction patterns of NiFe_2_O_4_ electrode annealed at different temperatures.

**Figure 2 nanomaterials-11-03238-f002:**
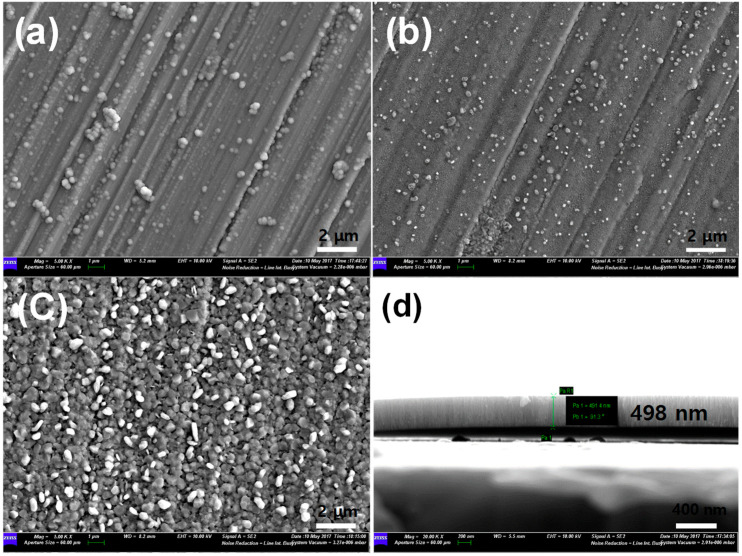
SEM images of NiFe_2_O_4_ electrode of (**a**) NFO-600, (**b**) NFO-700, and (**c**) NFO-800 °C. (**d**) Cross-sectional image of NiFe_2_O_4_ electrode on the glass.

**Figure 3 nanomaterials-11-03238-f003:**
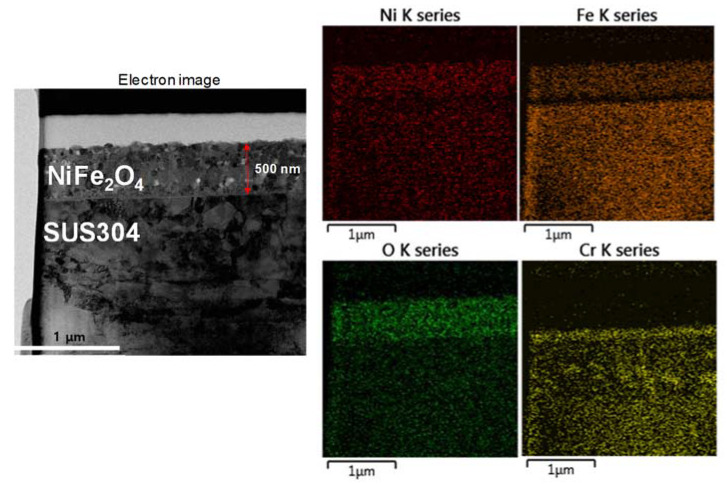
Cross-sectional and EDS mapping image of NFO-800 electrode.

**Figure 4 nanomaterials-11-03238-f004:**
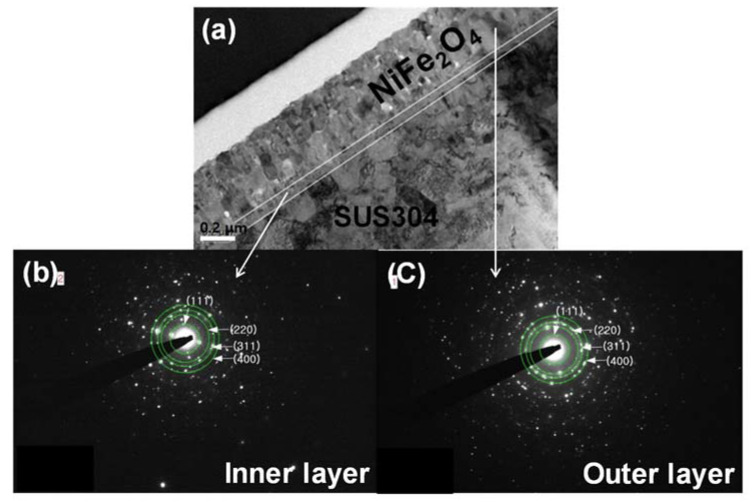
SAED pattern for the NFO-800 electrode. (**a**): cross-section; (**b**): Inner layer; (**c**): Outer layer.

**Figure 5 nanomaterials-11-03238-f005:**
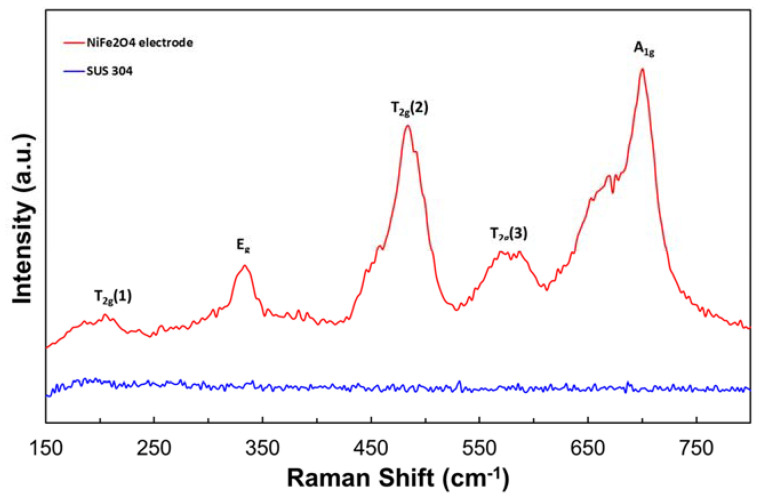
Raman spectra of NFO-800 and stainless steel.

**Figure 6 nanomaterials-11-03238-f006:**
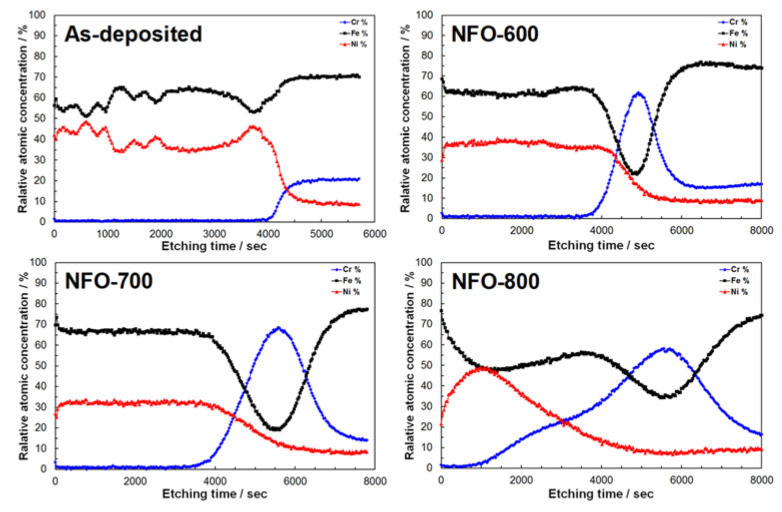
The “in depth” relative atomic concentrations of the NiFe_2_O_4_ electrode annealed at different temperatures from XPS-depth profiles.

**Figure 7 nanomaterials-11-03238-f007:**
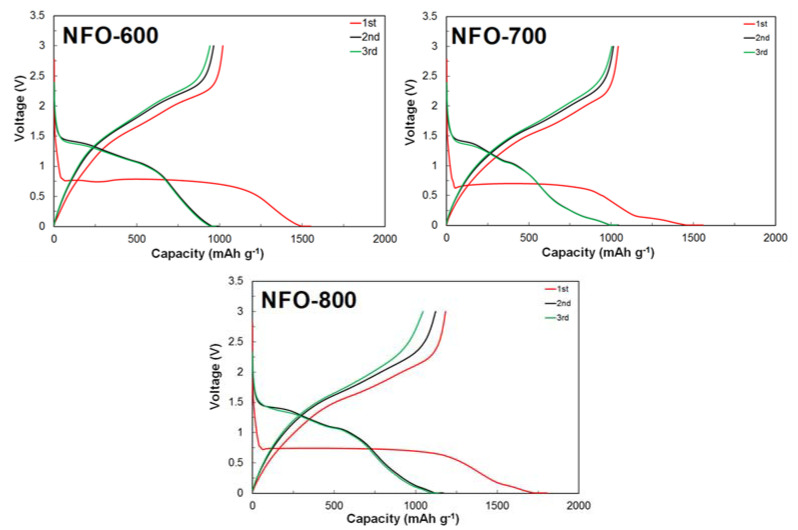
The initial charge-discharge profiles of the NiFe_2_O_4_ electrode annealed at different temperatures.

**Figure 8 nanomaterials-11-03238-f008:**
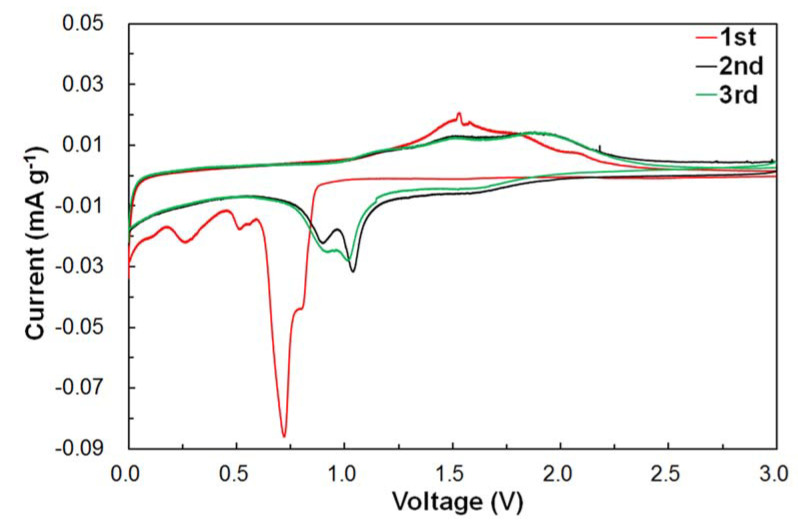
Cyclic voltammetry curves of the NiFe_2_O_4_ electrode annealed at 800 °C.

**Figure 9 nanomaterials-11-03238-f009:**
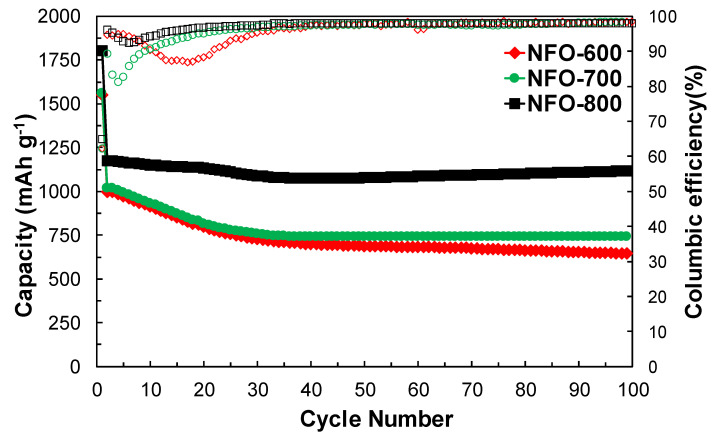
Cycle performance of the NiFe_2_O_4_ electrode annealed at different temperatures.

**Figure 10 nanomaterials-11-03238-f010:**
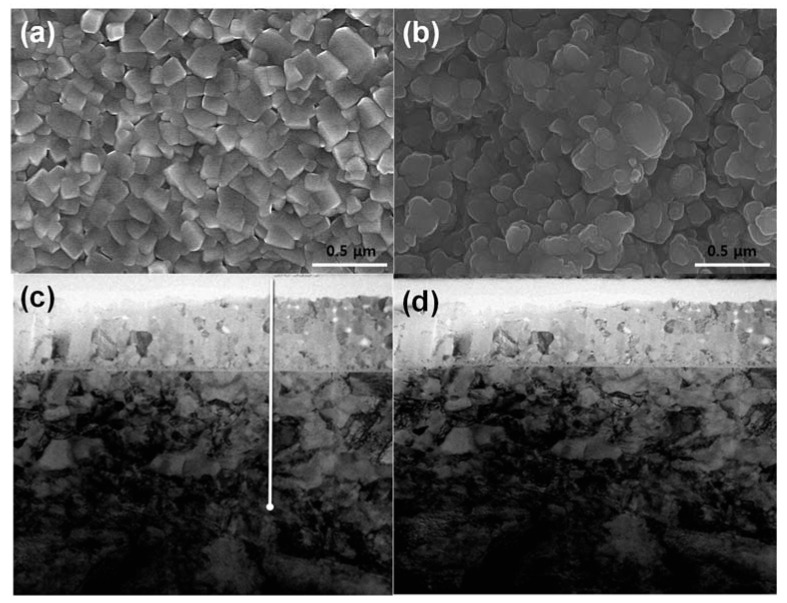
Surface morphologies of the NFO-800 electrode (**a**) before and (**b**) after cycles; (**c**) cross-section image; (**d**) cross-section image.

## Data Availability

The data presented in this study are available from the corresponding author upon request.

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
