# Peer review of "The Effect of Annealing Temperature on the Synthesis of Nickel Ferrite Films as High-Capacity Anode Materials for Lithium Ion Batteries"

_nanomaterials, 2021, doi:10.3390/nano11123238_

Round 1

Reviewer 1 Report

The effect of annealing temperature on performance of NiFe2O4 film anodes of LIB is investigated in this study. The English is poor, but the reviewer tried to improve it by having many comments on the English part along with scientific parts. Then the paper can be published after addressing following revisions:

  • The title is not more meaningful, and should be like: “The effect of annealing temperature on the synthesis of nickel ferrite films as high capacity anodes of lithium ion batteries”.
  • Abstract: “The anode materials providing high specific capacity with high cycling performance for is one of the key parameters for lithium ion batteries(LIBs). In this study, we investigated the effect of heat treatment of the NiFe2O4 film on microstructure and electrochemical properties for LIBs prepared by E-beam evaporation.” Is not meaningful, the authors talked about high capacity and long cycling life, but did not connected to their work, then the reviewer suggest to change it to “Anode materials providing high specific capacity with long cycling performance are key parameters for commercial lithium ion battery (LIB) applications. Herein, a high capacity NiFe2O4 (NFO) film anode is prepared by E-beam evaporation, and the effect of heat treatment is studied on microstructure and electrochemical properties of this anode.”
  • The reviewer Suggests to have some abbreviations and applied to the entire of manuscript for example: NiFe2O4 electrode annealed at 800 °C, NFO-800, and similar for annealed at 600 and 700 °C, NFO-600, NFO-700.
  • The rest of abstract should be changed to “The NiFe2O4 film annealed at 800 °C (NFO-800), showed a high crystallized structure and different surface morphologies compared to electrodes annealed at lower temperatures (NFO-600, NFO-700). High specific charge capacity (1100 mAh g-1) and a good capacity retention (95%) after 100 cycles are achieved by NFO-800 electrode. The main reason for good electrochemical performance of NFO-800 electrode is high structure integrity, which could improve the cycle stability with high charge capacity. The NiFe2O4 electrode with annealing process could be further proposed as alternative ferrite materials”
  • Keywords: please add “Anode” and “Thermal treatment”, these two are really keywords of this paper. if any limitation with the number of keywords, then please remove “Energy storage”.
  • Line 28-30: “For high performance of LIB, it is necessary to design the optimized anode materials with high capacity and cycle stability of electrode” should be “For having high performance LIBs, it is necessary to design optimized anode materials with high capacity and long cycle stability”.
  • Line 33: “The metal oxide, which is ferrites with general formula of an MFe2O4 (M= bivalent cation) spinel oxide,” should be “For instance, ferrites with general formula of MFe2O4 (M= bivalent cation) spinel oxide,”
  • Line 36: please add “(NFO)”, should be “…NiFe2O4 (NFO) is an inverse…” and also NiFe2O4 in line 38, 39, 44, 49, 55, and 59 in the introduction part. And please do it in the other sections as well.
  • Line 39: volume expansion (??%) [Ref??] the percentage should be added to the manuscript.
  • Line 41: when some words already abbreviated, the full name should not be used again. “it prevents their practical anode materials for Li-ion batteries” should be “it prevents their practical applications in LIBs”.
  • Line 42-43: “Many literatures have been reported regarding of the development the anode material with high specific capacity, rate capability, and cycle performance than graphite” should be “several studies discussed anode materials with high specific capacity, rate capability, and cycle performance than commercial graphite”
  • Line 47: “…surface area [ref??]”
  • Line 51: “...the deposition method…” please omit “the”
  • Line 53-54: “Further the LiMn2O4 thin film has been investigated the influence of annealing temperature on physical and electrochemical properties as anode materials [13].” Should be “Although, the LiMn2O4 (LMO) can be synthesised in room temperature in tetragonal [https://doi.org/10.1016/j.jpowsour.2018.11.074] and cubic [https://doi.org/10.1021/acssuschemeng.1c03747] forms., it can be synthesised at high temperatures, and the influence of annealing temperature on physical and electrochemical properties of LMO thin film is investigated as anode materials of LIBs [13].”
  • Line 58-59: “The effects of heat temperature of NiFe2O4 film were systematically investigated regarding of structural and electrochemical properties” should be “The effects of heat treatment on the synthesis of NFO film is systematically investigated, and structural and electrochemical properties will be discussed in details.”
  • Line 62-63: “NiFe2O4 film was prepared on stainless foil (SUS 304, thickness 25 μm) prepared by an electron beam evaporation system” should be “the NFO film coated on stainless foil (SUS 304, thickness 25 μm) using an electron beam evaporation system”
  • Line 65: “NiFe2O4 pellet (purity 99.9%)...” the supplier should be added “NFO pellet (purity 99.9%, from XXX)…”
  • Line 68: please omit “, respectively”. No need here.
  • Line 69-72: “The morphology of NiFe2O4 film surface were examined by a field emission scanning electron microscopy (S-4800, Hitachi). A high resolution transmission electron microscope (JEOL-2100F, HRTEM) was also used to capture the morphology and elemental mapping analysis was performed with energy dispersive spectroscopy of the NiFe2O4 sample” should be “The surface morphology of NFO film was examined by a field emission scanning electron microscopy, FESEM, (S-4800, Hitachi) and high resolution transmission electron microscope, HRTEM, (JEOL-2100F), elemental mapping analysis of the NFO sample also was performed with energy dispersive spectroscopy.”
  • Line 73-74: please add (FID-SEM) “…ion beam-scanning electron microscopy (FID-SEM) …”
  • Line 74-77: “The Pt coating was applied on the samples so as to prevent the morphological change by ion beam. The thin film X-ray diffraction (X-pert PRO MRD, Philips) was carried out Cu Kα radiation (λ = 1.5406 Å) operating at 40 76 kV and 30 mA at 2θ range from 10° and 90° at a scan rate of 0.01°” should be “To prevent the morphological change by ion beam, Pt coating was applied on the surface of electrodes. The X-ray diffraction (X-pert PRO MRD, Philips, Cu Kα, (λ = 1.5406 Å), operating at 40 kV and 30 mA) was carried at 2θ from 10° to 90° with a scan rate of 0.01°.”
  • Line 83-90: should be changed to “The electrochemical performance of the NFO anode electrodes were evaluated using 2032 coin cells assembled in a dry room. was used as electrode. The Li foil, and polypropylene 2400 (supplier??) were used as a counter and reference, and separator, respectively. The electrolyte was 1.3 M LiPF6 in ethylene carbonate (EC) and diethyl carbonate (DEC) (3:7 in volume) (supplier??). Galvanostatic charge-discharge and the cyclic voltammetry (CV) tests were performed in a potential range of 0.005-3.0 V (vs. Li+ /Li) with an applied current density of 0.1 C (1C=915 mA g-1 ) and at a scan rate of 0.5 mV s-1, respectively. All the electrochemical measurements were carried out at a room temperature using XXX??. The mass loading of NFO on the stainless steel according to the density of NFO, and average thickness of NFO, from cross section SEM images, is about XX?? ”.
  • Line 92-96: should be changed to “Obviously, the crystallinity of NFO film electrodes are depending on the heat treatment temperature, and they are shown in Fig.1, it can be seen with increasing of temperature from 600 to 800 °C the peaks become sharper, resulting in higher crystallinity of NFO materials. The crystallinity according to the Scherrer equation [Ref??] are calculated and are X?, Y?, and Z? nm for NFO-600, NFO-700, NFO-800 electrodes, respectively.”
  • 1, the sample names should be changed to NFO-600, NFO-700, and NFO-800 as well. What is the NiFe2O4-target? Do you mean the standard and commercial NiFe2O4? then the brand should be added to the experimental part, for example, “the NiFe2O4 from Aldrich, purity 99.9% is used as standard for comparison”. Then, line 108, “NiFe2O4 target” should be changed to “standard NFO”
  • Line 111-113: “The peak intensity become more strong according to the heat temperature. The peaks annealed at 800 °C revealed the highest sharp peak, indicating improvement of 112 crystallinity compared to that of lower temperature.” Should be changed to “The peak intensity become sharper when annealing temperature is increased from 600 to 800 °C, resulting high crystallinity for NFO-800 than NFO-600.”
  • Line 116-118: it is not necessary to say again the morphology characterization is done by SEM and HRTEM analysis, it is mentioned in the experimental part. Then “The morphology of the NiFe2O4 film was observed through the FE-SEM and HR-116 TEM. Fig. 2a-c shows surface morphologies of the NiFe2O4 electrode annealed at 600 °C, 117 700°C, and 800 °C.” should be changed to “Fig. 2a-c demonstrate surface morphologies of the all electrodes.”
  • Line 119: “…droplets is observed.” Should be “…droplets is also observed.”
  • Line 131: should be “shows”.
  • Line 140: what is “A”?
  • Fig 2: the scale bars should be clear and larger. The colour of (a) , (b), (c), and (d) labels should be same.
  • Line 144-145: the (d) should NFO-600 or NFO-700? It is not clear, and should be written “SEM images of (a) NFO-600, (b) NFO-700, and (c) NFO-800. (d) cross sectional image of NFO-???”
  • 3: what is the thickness of SUS304? Should be added to the cross section image.
  • Line 150: what do you mean by “The outer layer is grown by out diffusion of metal ions.” The sentence should be clear.
  • Line 207-210: should be “Fig. 7 shows the charge-discharge profiles of NFO electrodes at 0.1 C, all of electrodes showing typical voltage profiles of discharging(lithiation) and charging process(delithiation).”
  • Line 210-214: please add at least a reference for claims.
  • Line 217: omit “the subsequent cycle,”.
  • Line 218: “…that NiFe2O4 electrode becomes stable” should be “that NFO electrode becomes stable, and having reversible reactions”.
  • 9 should be changed to the charge capacities.
  • Line 258-261 are reported already in line 218-220, should be omitted. Instead the CE should be added to the fig. 9, and discussed here during cycling, then the difference between the electrodes can be seen by CE also.
  • Line 272 and 284: the electrode name should be added; it should be NFO-800?
  • 9: C rate performance (from 0.1C to 5C would be good to see) for all electrodes with corresponding CE should be added to this Figure as Fig. 9b.
  • Line 284: should be Figure 10.
  • 10: the before and after cycling cross section SEM images should be added to this figure.
  • A literature review about other reported NiFe2O4 electrodes should be added at the end of this section, maybe a table with about 10 references, and compare this study results with them vs. synthesis method, capacity, capacity retention, first CE, electrode density, mass loading, C rate performance.
  • Line 297-299: “In summary, the NiFe2O4 thin film electrodes have been successfully prepared via deposition method at room temperature. We investigated the annealing effects of NiFe2O4 electrode on the structural and electrochemical properties in Li-ion battery system.”, should be “In summary, the NFO thin film electrodes have been successfully prepared via deposition method, and the annealing effects on the structural and electrochemical properties of NFO electrodes were investigated in Li-ion battery system.”
  • Line 303: “current density of 0.1 Ahg-1” should be “current density of 91.5 mA g-1 (0.1C)”
  • Line 302-305: the first discharge capacity in anodes of LIBs are not important, how much charge is giving is much important. Then “The NiFe2O4 electrode annealed at 800 °C has shown a discharge capacity of 1804 mAh g-1 at a current density of 0.1 Ahg-1 and with stable cycle performance with high ratio than that of other annealing temperatures. A NiFe2O4 electrode also maintained their reversible capacity of 1100 mAh g-1 after 100 cycles.” should be ““The NFO-800 showing charge capacity about 1100 mAh g-1 and was stable after 100 cycles at 0.1C, 91.5 mA g-1, with a capacity retention of 95%.”
  • Line 307-309: “his section is not mandatory but can be added to the manuscript if the discussion is unusually long or complex” should be omitted.
  • Line 315: please omit “Please add:”

Author Response

The effect of annealing temperature on performance of NiFe2O4 film anodes of LIB is investigated in this study. The English is poor, but the reviewer tried to improve it by having many comments on the English part along with scientific parts.
 Then the paper can be published after addressing following revisions:
The title is not more meaningful, and should be like: “The effect of annealing temperature on the synthesis of nickel ferrite films as high capacity anodes of lithium ion batteries”.
→ the title has been changed as you pointed.
Abstract: “The anode materials providing high specific capacity with high cycling performance for is one of the key parameters for lithium ion batteries(LIBs). In this study, we investigated the effect of heat treatment of the NiFe2O4 film on microstructure and electrochemical properties for LIBs prepared by E-beam evaporation.” Is not meaningful, the authors talked about high capacity and long cycling life, but did not connected to their work, 
then the reviewer suggest to change it to “Anode materials providing high specific capacity with long cycling performance are key parameters for commercial lithium ion battery (LIB) applications. 
Herein, a high capacity NiFe2O4 (NFO) film anode is prepared by E-beam evaporation, and the effect of heat treatment is studied on microstructure and electrochemical properties of this anode.”
→ the abstract contents have been changed as you pointed.
The reviewer Suggests to have some abbreviations and applied to the entire of manuscript for example: NiFe2O4 electrode annealed at 800 °C, NFO-800, and similar for annealed at 600 and 700 °C, NFO-600, NFO-700. The rest of abstract should be changed to “The NiFe2O4 film annealed at 800 °C (NFO-800), showed a high crystallized structure and different surface morphologies compared to electrodes annealed at lower temperatures (NFO-600, NFO-700). High specific charge capacity (1100 mAh g-1) and a good capacity retention (95%) after 100 cycles are achieved by NFO-800 electrode. 
The main reason for good electrochemical performance of NFO-800 electrode is high structure integrity, which could improve the cycle stability with high charge capacity. 
The NiFe2O4 electrode with annealing process could be further proposed as alternative ferrite materials”
Keywords: please add “Anode” and “Thermal treatment”, these two are really keywords of this paper. if any limitation with the number of keywords, then please remove “Energy storage”.
→ the contents have been changed as you pointed.
Line 28-30: “For high performance of LIB, it is necessary to design the optimized anode materials with high capacity and cycle stability of electrode” should be “For having high performance LIBs,
 it is necessary to design optimized anode materials with high capacity and long cycle stability”.
→ the sentence has been changed as you pointed.
Line 33: “The metal oxide, which is ferrites with general formula of an MFe2O4 (M= bivalent cation) spinel oxide,” should be “For instance, ferrites with general formula of MFe2O4 (M= bivalent cation) spinel oxide,”
→ the sentence has been changed as you pointed.
Line 36: please add “(NFO)”, should be “…NiFe2O4 (NFO) is an inverse…” and also NiFe2O4 in line 38, 39, 44, 49, 55, and 59 in the introduction part. And please do it in the other sections as well.
→ I think, the NiFe2O4 will be better, because the NiFe2O4 without any treatment should be used. I have changed the NFO-600, 700, 800.
Line 39: volume expansion (??%) [Ref??] the percentage should be added to the manuscript.
→ I added the reference.
Line 41: when some words already abbreviated, the full name should not be used again. “it prevents their practical anode materials for Li-ion batteries” should be “it prevents their practical applications in LIBs”.
→ the sentence has been changed as you pointed.
Line 42-43: “Many literatures have been reported regarding of the development the anode material with high specific capacity, rate capability, and cycle performance than graphite” 
should be “several studies discussed anode materials with high specific capacity, rate capability, and cycle performance than commercial graphite”
→ the sentence has been changed as you pointed.
Line 47: “…surface area [ref??]”
→ I added the reference.
Line 51: “...the deposition method…” please omit “the”
→ I remove the “the”.
Line 53-54: “Further the LiMn2O4 thin film has been investigated the influence of annealing temperature on physical and electrochemical properties as anode materials [13].”
 Should be “Although, the LiMn2O4 (LMO) can be synthesised in room temperature in tetragonal [https://doi.org/10.1016/j.jpowsour.2018.11.074] and cubic [https://doi.org/10.1021/acssuschemeng.1c03747] forms., 
it can be synthesised at high temperatures, and the influence of annealing temperature on physical and electrochemical properties of LMO thin film is investigated as anode materials of LIBs [13].”
→ the sentence has been changed as you pointed.
Line 58-59: “The effects of heat temperature of NiFe2O4 film were systematically investigated regarding of structural and electrochemical properties” should be “The effects of heat treatment on the synthesis of  NFO film is systematically investigated, and structural and electrochemical properties will be discussed in details.”
→ the sentence has been changed as you pointed.
Line 62-63: “NiFe2O4 film was prepared on stainless foil (SUS 304, thickness 25 μm) prepared by an electron beam evaporation system” should be “the NFO film coated on stainless foil (SUS 304, thickness 25 μm) using an electron beam evaporation system”
Line 65: “NiFe2O4 pellet (purity 99.9%)...” the supplier should be added “NFO pellet (purity 99.9%, from XXX)…”
→ I added the company name.
Line 68: please omit “, respectively”. No need here.
→ I remove the “, respectively
Line 69-72: “The morphology of NiFe2O4 film surface were examined by a field emission scanning electron microscopy (S-4800, Hitachi). A high resolution transmission electron microscope (JEOL-2100F, HRTEM) 
was also used to capture the morphology and elemental mapping analysis was performed with energy dispersive spectroscopy of the NiFe2O4 sample” should be “The surface morphology of NFO film was examined by a field emission scanning electron microscopy, FESEM, (S-4800, Hitachi) and high resolution transmission electron microscope, HRTEM, (JEOL-2100F), elemental mapping analysis of the NFO sample also was performed with energy dispersive spectroscopy.”
→ the sentence has been changed as you pointed.
Line 73-74: please add (FID-SEM) “…ion beam-scanning electron microscopy (FID-SEM) …”
Line 74-77: “The Pt coating was applied on the samples so as to prevent the morphological change by ion beam. 
The thin film X-ray diffraction (X-pert PRO MRD, Philips) was carried out Cu Kα radiation (λ = 1.5406 Å) operating at 40 76 kV and 30 mA at 2θ range from 10° and 90° at a scan rate of 0.01°” should be
 “To prevent the morphological change by ion beam, Pt coating was applied on the surface of electrodes. The X-ray diffraction (X-pert PRO MRD, Philips, Cu Kα, (λ = 1.5406 Å), operating at 40 kV and 30 mA) 
was carried at 2θ from 10° to 90° with a scan rate of 0.01°.”
Line 83-90: should be changed to “The electrochemical performance of the NFO anode electrodes were evaluated using 2032 coin cells assembled in a dry room. was used as electrode. 
The Li foil, and polypropylene 2400 (supplier??) were used as a counter and reference, and separator, respectively. The electrolyte was 1.3 M LiPF6 in ethylene carbonate (EC) and diethyl carbonate (DEC) (3:7 in volume) (supplier??). 
Galvanostatic charge-discharge and the cyclic voltammetry (CV) tests were performed in a potential range of 0.005-3.0 V (vs. Li+ /Li) with an applied current density of 0.1 C (1C=915 mA g-1 ) and at a scan rate of 0.5 mV s-1, respectively. 
All the electrochemical measurements were carried out at a room temperature using XXX??. 
The mass loading of NFO on the stainless steel according to the density of NFO, and average thickness of NFO, from cross section SEM images, is about XX?? ”.
Line 92-96: should be changed to “Obviously, the crystallinity of NFO film electrodes are depending on the heat treatment temperature, and they are shown in Fig.1,
 it can be seen with increasing of temperature from 600 to 800 °C the peaks become sharper, resulting in higher crystallinity of NFO materials. 
The crystallinity according to the Scherrer equation [Ref??] are calculated and are X?, Y?, and Z? nm for NFO-600, NFO-700, NFO-800 electrodes, respectively.” 1, the sample names should be changed to NFO-600, NFO-700, and NFO-800 as well. 
What is the NiFe2O4-target? Do you mean the standard and commercial NiFe2O4? then the brand should be added to the experimental part, for example, 
“the NiFe2O4 from Aldrich, purity 99.9% is used as standard for comparison”. Then, line 108, “NiFe2O4 target” should be changed to “standard NFO”
Line 111-113: “The peak intensity become more strong according to the heat temperature. The peaks annealed at 800 °C revealed the highest sharp peak, indicating improvement of 112 crystallinity compared to that of lower temperature.” Should be changed to “The peak intensity become sharper when annealing temperature is increased from 600 to 800 °C, resulting high crystallinity for NFO-800 than NFO-600.”
→ the sentence has been changed as you pointed.
Line 116-118: it is not necessary to say again the morphology characterization is done by SEM and HRTEM analysis, it is mentioned in the experimental part. 
→ the sentence has been removed as you pointed
Then “The morphology of the NiFe2O4 film was observed through the FE-SEM and HR-116 TEM. Fig. 2a-c shows surface morphologies of the NiFe2O4 electrode annealed at 600 °C, 117 700°C, and 800 °C.” 
should be changed to “Fig. 2a-c demonstrate surface morphologies of the all electrodes.”
Line 119: “…droplets is observed.” Should be “…droplets is also observed.”
→ the sentence has been changed as you pointed.
Line 131: should be “shows”.
Line 140: what is “A”?
→ the sentence has been changed A to “the”.
Fig 2: the scale bars should be clear and larger. The colour of (a) , (b), (c), and (d) labels should be same.
→ the scale bars are added in the photos.
Line 144-145: the (d) should NFO-600 or NFO-700? It is not clear, and should be written “SEM images of (a) NFO-600, (b) NFO-700, and (c) NFO-800. (d) cross sectional image of NFO-???”
3: what is the thickness of SUS304? Should be added to the cross section image.
→ the substrate in Fig. 2d is glass. I added it in the caption of Fig.2.
Line 150: what do you mean by “The outer layer is grown by out diffusion of metal ions.” The sentence should be clear.
→ the sentence is changed by “The outer layer is grown by out diffusion of metal ions from the substrate.”
Line 207-210: should be “Fig. 7 shows the charge-discharge profiles of NFO electrodes at 0.1 C, all of electrodes showing typical voltage profiles of discharging(lithiation) and charging process(delithiation).”
→ the sentence has been changed as you pointed out.
Line 210-214: please add at least a reference for claims.
Line 217: omit “the subsequent cycle,”.
Line 218: “…that NiFe2O4 electrode becomes stable” should be “that NFO electrode becomes stable, and having reversible reactions”. 9 should be changed to the charge capacities.
→ the sentence has been changed as you pointed out.
Line 258-261 are reported already in line 218-220, should be omitted. Instead the CE should be added to the fig. 9, and discussed here during cycling, then the difference between the electrodes can be seen by CE also.
Line 272 and 284: the electrode name should be added; it should be NFO-800?
→ the sentence has been changed as you pointed out.
9: C rate performance (from 0.1C to 5C would be good to see) for all electrodes with corresponding CE should be added to this Figure as Fig. 9b.
→ the C rate performance will be carried out in further study and the CE for cycle performance of 3 electrodes was almost same. 
Line 284: should be Figure 10.
→ I revised the Fig 10.
10: the before and after cycling cross section SEM images should be added to this figure.
→ we did not observe the SEM image of electrode before and after cycling. We assumed that the electrode may not be changed during cycling.
A literature review about other reported NiFe2O4 electrodes should be added at the end of this section, maybe a table with about 10 references, and compare this study results with them vs. synthesis method, capacity, capacity retention, first CE, electrode density, mass loading, C rate performance.
→ Our study is little difference with other works. We have investigated the effect of annealing temperature in NiFe2O4 electrode by E-beam method. This is first work for synthesis method. We will do further study regarding of mechanisms for high electrochemical properties in later.
Line 297-299: “In summary, the NiFe2O4 thin film electrodes have been successfully prepared via deposition method at room temperature. 
We investigated the annealing effects of NiFe2O4 electrode on the structural and electrochemical properties in Li-ion battery system.”, 
should be “In summary, the NFO thin film electrodes have been successfully prepared via deposition method, and the annealing effects on the structural and electrochemical properties of NFO electrodes were investigated in Li-ion battery system.”
→ the sentence has been changed as you pointed out.
Line 303: “current density of 0.1 Ahg-1” should be “current density of 91.5 mA g-1 (0.1C)”
→ the sentence has been changed as you pointed out.
Line 302-305: the first discharge capacity in anodes of LIBs are not important, how much charge is giving is much important. 
Then “The NiFe2O4 electrode annealed at 800 °C has shown a discharge capacity of 1804 mAh g-1 at a current density of 0.1 Ahg-1 and with stable cycle performance with high ratio than that of other annealing temperatures. A NiFe2O4 electrode also maintained their reversible capacity of 1100 mAh g-1 after 100 cycles.” should be ““The NFO-800 showing charge capacity about 1100 mAh g-1 and was stable after 100 cycles at 0.1C, 91.5 mA g-1, with a capacity retention of 95%.”
→ the sentence has been changed as you pointed out.
Line 307-309: “his section is not mandatory but can be added to the manuscript if the discussion is unusually long or complex” should be omitted.
Line 315: please omit “Please add:”
→ the above sentences hav been changed as you pointed out

Reviewer 2 Report

  1. Milan et al. described the preparation and electrochemical performances of NiFe2O4 film as anodes electrode for lithium ion battery. The effects of the heat treatment temperature on the structure and electrochemical were clearly systematically investigated. Before consideration for publication, however, the authors still need to address some major concerns and make some necessary revisions. The following comments and suggestions are provided for the authors.

    1. In Fig. 1, the XRD pattern of the stainless substrate is suggested to provide.
    2. In Fig. 2, the authors claimed that “the heat treatment of stainless steel changes their chemical composition and the thickness of the passive film”. The morphologies of the stainless steel treated at different temperature should be discussed.
    3. In Fig. 9, please also give the areal capacities and Columbic efficiencies of the three electrodes.
    4. The authors stated that the heat treatment maintained the structure integrity of NiFe2O4 and thus improved the cycle stability. How about the interlayer between the NiFe2O4 and substrate?

Author Response

1.Milan et al. described the preparation and electrochemical performances of NiFe2O4 film as anodes electrode for lithium ion battery. 
The effects of the heat treatment temperature on the structure and electrochemical were clearly systematically investigated. Before consideration for publication, 
however, the authors still need to address some major concerns and make some necessary revisions. The following comments and suggestions are provided for the authors.
1.In Fig. 1, the XRD pattern of the stainless substrate is suggested to provide.
→ the peak of SUS shows 1 peak in XRD. So we have shown the star on Fig.1. 
2.In Fig. 2, the authors claimed that “the heat treatment of stainless steel changes their chemical composition and the thickness of the passive film”. The morphologies of the stainless steel treated at different temperature should be discussed.
→ we have assumed the above mechanism from other references. And the electrode chemical effect can be from only NiFe2O4 electrodes. We will do further study between stainless steel and NiFe2O4 electrode. At that time, we will observe the morphologies of SUS after heat treatment.
3.In Fig. 9, please also give the areal capacities and Columbic efficiencies of the three electrodes.
→ the sentence has been changed as you pointed out
4.The authors stated that the heat treatment maintained the structure integrity of NiFe2O4 and thus improved the cycle stability. How about the interlayer between the NiFe2O4 and substrate?
→ we have guessed that the interlayer may play a role for capacity improvement. However, we did not investigate in this study, we will do it in later.

Round 2

Reviewer 1 Report

The authors replied to about 50% of comments raised by the reviewer, however many of them are missing and should be addressed before publishing the paper. The reviewer spent a lot of time for the first revision and also now for the second revision “to improve the quality of the paper”, so please do the following comments patiently before submitting the second version to the journal:

  • Title: “film” should be “films”
  • Line 31: “cycle stability of electrode” should be “long cycle stability”.
  • Line 55: the following references should be added to the manuscript: “tetragonal [https://doi.org/10.1016/j.jpowsour.2018.11.074]” and “cubic [https://doi.org/10.1021/acssuschemeng.1c03747]”
  • Line 65-66: this is not a correct sentence, two times prepared is used: “NiFe2O4 film was prepared on stainless foil (SUS 304, thickness 25 μm) prepared by an electron beam evaporation system” should be “the NFO film coated on stainless foil (SUS 304, thickness 25 μm) using an electron beam evaporation system”
  • Line 76: please add (FID-SEM) “…ion beam-scanning electron microscopy (FID-SEM) …”
  • Line 77-80: “The Pt coating was applied on the samples so as to prevent the morphological change by ion beam. The thin film X-ray diffraction (X-pert PRO MRD, Philips) was carried out Cu Kα radiation (λ = 1.5406 Å) operating at 40 76 kV and 30 mA at 2θ range from 10° and 90° at a scan rate of 0.01°” should be “To prevent the morphological change by ion beam, Pt coating was applied on the surface of electrodes. The X-ray diffraction (X-pert PRO MRD, Philips, Cu Kα, (λ = 1.5406 Å), operating at 40 kV and 30 mA) was carried at 2θ from 10° to 90° with a scan rate of 0.01°.”
  • Line 85-92: should be changed to “The electrochemical performance of the NFO anode electrodes were evaluated using 2032 coin cells assembled in a dry room. was used as electrode. The Li foil, and polypropylene 2400 (supplier??) were used as a counter and reference, and separator, respectively. The electrolyte was 1.3 M LiPF6 in ethylene carbonate (EC) and diethyl carbonate (DEC) (3:7 in volume) (supplier??). Galvanostatic charge-discharge and the cyclic voltammetry (CV) tests were performed in a potential range of 0.005-3.0 V (vs. Li+ /Li) with an applied current density of 0.1 C (1C=915 mA g-1 ) and at a scan rate of 0.5 mV s-1, respectively. All the electrochemical measurements were carried out at a room temperature using XXX?? (please add the instrument name, for example biologic). The mass loading of NFO on the stainless steel according to the density of NFO, and average thickness of NFO, from cross section SEM images, is about XX?? ”.
  • Line 94-97: should be changed to “Obviously, the crystallinity of NFO film electrodes are depending on the heat treatment temperature, and they are shown in Fig.1, it can be seen with increasing of temperature from 600 to 800 °C the peaks become sharper, resulting in higher crystallinity of NFO materials.
  • At the end of line 97, the following sentence should be added and also is important to know about the crystallinity of sample with different heat treatments, should be calculated. “The crystallinity according to the Scherrer equation [Ref??] are calculated and are X?, Y?, and Z? nm for NFO-600, NFO-700, NFO-800 electrodes, respectively.”
  • 1: what do you mean by “NFO”, Do you mean the standard and commercial NiFe2O4? then the brand should be added to the experimental part, for example, “the NiFe2O4 from Aldrich, purity 99.9% is used as standard for comparison”. Then, line 101, “NiFe2O4 target” should be changed to “standard NFO”.
  • Line 105: …resulting “in”.
  • Line 123: should be “shows”.
  • Line 134-135: the (d) should be for NFO-600 or NFO-700? It is not clear, and should be written “(d) cross sectional image of NFO-???”
  • 3: what is the thickness of SUS304? Should be added to the cross section image.
  • 4: the scale bare and the number is not clear for fig. 4a.
  • Line 194-195: should be “Fig. 7 shows the charge-discharge profiles of NFO electrodes at 0.1 C, all of electrodes showing typical voltage profiles of discharging(lithiation) and charging process(delithiation).”
  • Line 197-200: please add at least a reference for claims.
  • Line 203: omit “the subsequent cycle,”.
  • Line 205: “…that NiFe2O4 electrode becomes stable” should be “that NFO electrode becomes stable, and having reversible reactions”.
  • 9 should be changed to the charge capacities not discharge capacities, or add charge capacities also to this figure.
  • Line 245-247 are reported already in lines 205-207, should be omitted. Instead the CE should be added to the fig. 9, and discussed here during cycling, then the difference between the electrodes can be seen by CE also.
  • Line 258: the electrode name should be added; it should be NFO-800?
  • C rate performance (from 0.1C to 5C would be good to see) for all electrodes with corresponding CE should be added to the Fig. 9 as Fig. 9b. this test is very common test for every battery study and authors should added to the manuscript. The same CE also is OK, please just reported in the manuscript and figures.
  •  
  • Regarding Fig. 10: why the authors assuming that the electrode may not be changed during cycling? The reason with a evidence should be added to the manuscript, or the before and after cycling cross section SEM images should be added to this figure.
  • This could improve the paper quality, so the reviewer suggests to add it: A literature review about other reported NiFe2O4 electrodes should be added at the end of this section, maybe a table with about 10 references, and compare this study results with them vs. synthesis method, capacity, capacity retention, first CE, electrode density, mass loading, C rate performance.
  • Line 290: “current density of 0.1 mAh g-1” should be “current density of 91.5 mA g-1 (0.1C)”. “dimension for capacity: mAh g-1”, and “dimension for the current density: mA g-1”

Author Response

The authors replied to about 50% of comments raised by the reviewer, however many of them are missing and should be addressed before publishing the paper. The reviewer spent a lot of time for the first revision and also now for the second revision “to improve the quality of the paper”, so please do the following comments patiently before submitting the second version to the journal:
→ Thank you for your devoted effort to review my paper. First of all, I am sorry for the shortage of reviewer reply. However, I have tried to make up my revision paper following the reviewer comments. So I revise my manuscript again as the reviewer pointed out. 
Title: “film” should be “films”
→ the title is changed as you pointed out.
Line 31: “cycle stability of electrode” should be “long cycle stability”.
→ the word is changed as you pointed out.
Line 55: the following references should be added to the manuscript: “tetragonal [https://doi.org/10.1016/j.jpowsour.2018.11.074]” and “cubic [https://doi.org/10.1021/acssuschemeng.1c03747]”
→ 2 reference papers were added in the manuscript.
Line 65-66: this is not a correct sentence, two times prepared is used: “NiFe2O4 film was prepared on stainless foil (SUS 304, thickness 25 μm) prepared by an electron beam evaporation system” should be “the NFO film coated on stainless foil (SUS 304, thickness 25 μm) using an electron beam evaporation system”
→ the word is changed as you pointed out.
Line 76: please add (FID-SEM) “…ion beam-scanning electron microscopy (FID-SEM) …”
→ the word is changed as you pointed out.
Line 77-80: “The Pt coating was applied on the samples so as to prevent the morphological change by ion beam. The thin film X-ray diffraction (X-pert PRO MRD, Philips) was carried out Cu Kα radiation (λ = 1.5406 Å) operating at 40 76 kV and 30 mA at 2θ range from 10° and 90° at a scan rate of 0.01°” should be “To prevent the morphological change by ion beam, Pt coating was applied on the surface of electrodes. The X-ray diffraction (X-pert PRO MRD, Philips, Cu Kα, (λ = 1.5406 Å), operating at 40 kV and 30 mA) was carried at 2θ from 10° to 90° with a scan rate of 0.01°.”
→ the sentences were changed as you pointed out.
Line 85-92: should be changed to “The electrochemical performance of the NFO anode electrodes were evaluated using 2032 coin cells assembled in a dry room. was used as electrode. The Li foil, and polypropylene 2400 (supplier??) were used as a counter and reference, and separator, respectively. The electrolyte was 1.3 M LiPF6 in ethylene carbonate (EC) and diethyl carbonate (DEC) (3:7 in volume) (supplier??). Galvanostatic charge-discharge and the cyclic voltammetry (CV) tests were performed in a potential range of 0.005-3.0 V (vs. Li+ /Li) with an applied current density of 0.1 C (1C=915 mA g-1 ) and at a scan rate of 0.5 mV s-1, respectively. All the electrochemical measurements were carried out at a room temperature using XXX?? (please add the instrument name, for example biologic). The mass loading of NFO on the stainless steel according to the density of NFO, and average thickness of NFO, from cross section SEM images, is about XX?? ”.
→ the sentences were changed as you pointed out.
Line 94-97: should be changed to “Obviously, the crystallinity of NFO film electrodes are depending on the heat treatment temperature, and they are shown in Fig.1, it can be seen with increasing of temperature from 600 to 800 °C the peaks become sharper, resulting in higher crystallinity of NFO materials.
→ the sentences were changed as you pointed out.
At the end of line 97, the following sentence should be added and also is important to know about the crystallinity of sample with different heat treatments, should be calculated. “The crystallinity according to the Scherrer equation [Ref??] are calculated and are X?, Y?, and Z? nm for NFO-600, NFO-700, NFO-800 electrodes, respectively.”
→ I could not understand how do you calculate the crystallinity? Is it possible to show the number of calculation? I know the Scherrer equation which can calculate crystal size of the particles.
1: what do you mean by “NFO”, Do you mean the standard and commercial NiFe2O4? then the brand should be added to the experimental part, for example, “the NiFe2O4 from Aldrich, purity 99.9% is used as standard for comparison”. Then, line 101, “NiFe2O4 target” should be changed to “standard NFO”.
→ the sample name was changed as you pointed out.
Line 105: …resulting “in”.
→ the word is added as you pointed out.
Line 123: should be “shows”.
→ the word is added as you pointed out.

Line 134-135: the (d) should be for NFO-600 or NFO-700? It is not clear, and should be written “(d) cross sectional image of NFO-???”
→ the Fig 2d is the NiFe2O4 on the glass, which is coated in E-beam without annealing. From the Fig 2d we know the thickness of NiFe2O4.
3: what is the thickness of SUS304? Should be added to the cross section image.
→ the thickness of SUS was in the experimental section(25 μm). And the part of cross-section image also is shown in Fig 3.
4: the scale bare and the number is not clear for fig. 4a.
→ the scale bar is added in Fig. 4a.
Line 194-195: should be “Fig. 7 shows the charge-discharge profiles of NFO electrodes at 0.1 C, all of electrodes showing typical voltage profiles of discharging(lithiation) and charging process(delithiation).”
→ the sentences were changed as you pointed out.
Line 197-200: please add at least a reference for claims.
→ the reference was added.
Line 203: omit “the subsequent cycle,”.
→ “the subsequent cycle,” was removed.
Line 205: “…that NiFe2O4 electrode becomes stable” should be “that NFO electrode becomes stable, and having reversible reactions”.
→ the sentences were changed as you pointed out.
9 should be changed to the charge capacities not discharge capacities, or add charge capacities also to this figure.
→ I do not understand what 9? Do you mean Fig. 9? 
Line 245-247 are reported already in lines 205-207, should be omitted. Instead the CE should be added to the fig. 9, and discussed here during cycling, then the difference between the electrodes can be seen by CE also.
→ the first charge-discharge capacities of electrodes are different. Even though the value is almost same, we got the data from separated electrode for charge-discharge and cycle performance. And the CE was added in the Fig. 9.
Line 258: the electrode name should be added; it should be NFO-800?
→ the word was added in the sentence.
C rate performance (from 0.1C to 5C would be good to see) for all electrodes with corresponding CE should be added to the Fig. 9 as Fig. 9b. this test is very common test for every battery study and authors should added to the manuscript. The same CE also is OK, please just reported in the manuscript and figures.
→ In this study, we have focused on the effects of annealing temperature in charge-discharge and cycle stability. We did not implement the C-rate test. We will do next study. 
Regarding Fig. 10: why the authors assuming that the electrode may not be changed during cycling? The reason with a evidence should be added to the manuscript, or the before and after cycling cross section SEM images should be added to this figure.
→ I added the TEM cross-section image of NFO-800 before and after cycling.
This could improve the paper quality, so the reviewer suggests to add it: A literature review about other reported NiFe2O4 electrodes should be added at the end of this section, maybe a table with about 10 references, and compare this study results with them vs. synthesis method, capacity, capacity retention, first CE, electrode density, mass loading, C rate performance.
→ In this study, we firstly reported the effects of annealing of NiFe2O¬4 electrode on stainless steel. The comparison with other reports are not meaningful. Because the most paper is written by general Li-ion system with Cu current collector. This is totally different with other paper and these things are main subject of our study. In my knowledge, there is no reports regarding of Cu and electrode materials. After annealing, there are unique properties between SUS and NiFe2O4 electrode. We mainly find the phenomena in this study. 
Line 290: “current density of 0.1 mAh g-1” should be “current density of 91.5 mA g-1 (0.1C)”. “dimension for capacity: mAh g-1”, and “dimension for the current density: mA g-1”
→ the word(current density of 91.5 mA g-1 (0.1C)) was added in the sentence

Reviewer 2 Report

All issues are well addressed.

The ms can be accepted in present form.

Author Response

Thank you fot your kind review.